# Improved Electrochemical Properties of an Ni-Based YSZ Cermet Anode for the Direct Supply of Methane by Co Alloying with an Impregnation Method

**Nicharee Wongsawatgul [1,2]**, **Soamwadee Chaianansutcharit [3]**, **Kazuhiro Yamamoto [4]**,
**Makoto Nanko [5] and Kazunori Sato [4,6],***

[1]   Program of Materials Science, Nagaoka University of Technology, Nagaoka, Niigata 940-2188, Japan;
     nicharee.w@gmail.com
[2]   Program of Petrochemistry, Faculty of Science, Chulalongkorn University, Bangkok 10330, Thailand
[3]   Department of Chemistry, Faculty of Science, Chulalongkorn University, Bangkok 10330, Thailand;
     soamwadee.c@chula.ac.th
[4]   Department of Materials Science and Technology, Nagaoka University of Technology, Nagaoka,
     Niigata 940-2188, Japan; kyamamoto@vos.nagaokaut.ac.jp
[5]   Department of Mechanical Engineering, Nagaoka University of Technology, Nagaoka, Niigata 940-2188,
     Japan; nanko@mech.nagaokaut.ac.jp
[6]   Advanced Methane-Utilization Research Center, Nagaoka, Niigata 940-2188, Japan
*   Correspondence: sato@vos.nagaokaut.ac.jp

**Abstract:** To avoid the proneness to degradation due to coking in the operation of solid oxide fuel cells (SOFCs) directly running on methane ($CH_4$) fuels, a modified porous anode of the $Ni_{1-X}Co_X$/YSZ (yttria-stabilized zirconia) cermet prepared by an impregnation method is presented. The influence of the Co alloying content on the cermet microstructure, SOFC characteristics, and prolonged cell performance stability has been studied. Co was incorporated into Ni and formed a solid solution of $Ni_{1-X}Co_X$ alloy connected with the YSZ as the cermet anode. The porous microstructure of the $Ni_{1-X}Co_X$/YSZ cermet anode formed by sintering exhibited a grain growth with an increase in the Co alloying content. The electrochemical performance of the cells consisting of the $Ni_{1-X}Co_X$/YSZ cermet anode, the YSZ electrolyte, and the LSM ($La_{0.8}Sr_{0.2}MnO_3$) cathode showed an enhancement by the $Ni_{1-X}Co_X$ impregnation treatment for the respective supply of $H_2$ and $CH_4$ to the anode. The cell using the $Ni_{0.75}Co_{0.25}$/YSZ cermet anode (the $Ni_{0.75}Co_{0.25}$ cell) showed the highest cell performance among the cells tested. In particular, the performance enhancement of this cell was found to be more significant for $CH_4$ than that for $H_2$; a 45% increase in the maximum power density for $CH_4$ and a 17% increase for $H_2$ at 750 °C compared with the performance of the cell using the Ni/YSZ cermet anode. Furthermore, the prolonged cell performance stability with a continuous $CH_4$ supply was found for the $Ni_{0.85}Co_{0.15}$ and $Ni_{0.75}Co_{0.25}$ cells at least for 60 h at 750 °C. These enhancement effects were caused by the optimum porous microstructure of the cermet anode with the low anodic polarization resistance.

**Keywords:** cermet; nickel–cobalt alloy; yttria-stabilized zirconia; impregnation; anode; solid oxide fuel cell; methane

## 1. Introduction

Solid oxide fuel cells (SOFCs) have been attracting much interest as a role of the alternative electric power supply system. The main feature of SOFCs is high power generation efficiency due to the direct conversion of chemical energy into electricity, which is not limited by the Carnot efficiency, and thus

SOFCs offer considerably higher power generation efficiencies than conventional energy conversion systems. SOFCs are usually operated at temperatures higher than about 600 °C, and they are able to operate with both hydrogen ($H_2$) and reformed hydrocarbon (HC) or ethanol fuels. Although the chemical reaction with $H_2$ generates only water as the by-product, storage problems for $H_2$ have not been practically solved so far. The direct use of gaseous HCs for SOFCs has therefore been recently studied [1–6]. One of the most beneficial alternative fuels for SOFCs is $CH_4$, which is the major component of natural gas, coal-bed gas, and biogas [7–10]. The highest hydrogen-to-carbon molar ratio of $CH_4$ among HCs can reduce the emission of $CO_2$ after the combustion.

Electrochemical oxidation of fuels in SOFCs occurs at the anode. The conventional anode is a cermet made of Ni and a yttria-stabilized zirconia (YSZ) frame, which is processed by sintering a mixture of starting NiO and YSZ powders with a post-reduction by $H_2$. The Ni/YSZ anode works well for $H_2$ as a fuel due to its high electrochemical catalytic activity for the oxidation of $H_2$. However, the coarsening of Ni particles occurring at high operating temperatures results in a decrease in the triple phase boundary (TPB) and reduce the long-term stability of the cell performance.

Moreover, the Ni/YSZ cermet anode suffers from limitations when HCs are directly used; the steam reforming of HCs into $H_2$-rich fuels is necessary. Improvement in the efficiency of the Ni/YSZ cermet anode running on HCs has been frequently studied. Since Ni has a high catalytic activity for thermal C–H bond breaking in HC molecules, the carbon produced by thermal dissociation can be oxidized to form CO. However, the excessively formed carbon atoms are not gasified and left on the surface of Ni particles, which are polymerized to form graphite and are deposited on the surface of Ni particles, resulting in the deactivation of the anode [11].

Humidified $H_2$ can alleviate the carbon deposition at the Ni/YSZ cermet anode [12]; however, the sintering of Ni particles is facilitated and the cell is deactivated by high humidity content and the cell system needs to be more complex [13]. The alloying of Ni with other metallic components can be an alternative way to reduce the carbon deposition. Although the incorporation of Cu into Ni was found to decrease the carbon deposition from $CH_4$ [14,15], the SOFC operating temperatures are unfavorably limited due to a requirement of low sintering temperatures for the Cu–Ni alloy. Moreover, the Cu diffusion into the YSZ frame causes a degradation of the Ni–Cu/YSZ cermet [16–18]. The alloying of Sn into Ni has been studied for the carbon formation retardation due to the preferential oxidation of the carbon atoms rather than formation of C–C bond; however, the segregation of Sn to the surface of the Ni particle occupies the electrochemically active site and increases the polarization resistance of the cell [19]. The alloying of Ni with Co is interesting since Co has a similar metallic property to Ni [20]. A small amount of the Co incorporation restrained the Ni particle coarsening and promoted the cell performance for $H_2$ [21]. The Co alloying also improved the electrocatalytic activity and stability for $H_2$ containing CO or $H_2S$ as a fuel [22]. The addition of Co to the Ni/GDC (gadolinia-doped ceria) cermet anode decreased the anodic polarization resistance and moderated the carbon deposition [23]. Nevertheless, the proper mechanism of inhibiting the carbon deposition by Co alloying to Ni in the cermet anode seems to be still unclear and the electrochemical performance and the prolonged stability has not been carefully studied. The objective of this work is to investigate the influence of Co alloying in the Ni/YSZ cermet in terms of morphology and the electrochemical performance, particularly the prolonged stability of the cell for a dry $CH_4$ supply. The thermal dissociation of $CH_4$ molecules at the $Ni_{1-X}Co_X$/YSZ (X = 0–0.5) cermet anode has been investigated by using symmetric cells to elucidate the enhancement effect on the electrocatalytic activity for the electrochemical oxidation of $CH_4$.

## 2. Materials and Methods

The $Ni_{1-X}Co_X$/YSZ (x = 0, 0.05, 0.15, 0.25, 0.50) cermet powders were prepared by the impregnation method combined with an $H_2$-reducing treatment after sintering. The first step in preparing the powders was to use appropriate amounts of $Ni(NO_3)_2 \cdot 6H_2O$ (98% purity, Nacalai Tesque, Kyoto, Japan) and $Co(NO_3)_3 \cdot 9H_2O$ (98% purity, Nacalai Tesque) dissolved in distilled water, and subsequently the YSZ powder (8 mol%$Y_2O_3$-$ZrO_2$(TZ-8Y), Tosoh, Tokyo, Japan) was proportionately added into the

solution with the composition of 60% $Ni_{1-X}Co_X$ + 40%YSZ (wt %) under continuously vigorous stirring at room temperature. Each suspension was stirred for 2 h and the temperature was subsequently raised up to 180 °C for dehydration. The dehydrated powder was calcined at 800 °C for 5 h in air.

The calcined powder was manually ground in an agate mortar with a pestle for 1 h in ethanol, and the dried ground powder was mixed with glycerol to prepare slurries. The slurry was painted as the anode at one face of the YSZ electrolyte disk (15 mm diameter, 0.23 ± 0.01 mm thick and surface polished, Nikkato, Osaka, Japan). The slurry-painted YSZ disk was heated at 1300 °C for 3 h in air. The $La_{0.8}Sr_{0.2}MnO_3$–YSZ composite was used as the cathode material; it was prepared by painting a slurry of the 70%$La_{0.8}Sr_{0.2}MnO_3$ (LSM, Seimi Chemical, Chigasaki, Kanagawa Prefecture, Japan) + 30%YSZ (wt %) powder mixture with glycerol to the other face of the YSZ disk and heated at 1200 °C for 3 h in air to form the $La_{0.8}Sr_{0.2}MnO_3$–YSZ (denoted as LSM–YSZ) cathode. The geometrical area of both the anode and cathode was 0.28 $cm^2$ with approximately 70 μm thick. As the reference electrode a Pt paste (TR-7603, Tanaka Kikinzoku Kogyo, Tokyo, Japan) was painted as a small circle close to the anode with a small gap at least 2 mm away from the edge of the anode circle. A circle with a 3 mm diameter of Pt mesh (# 100 mesh), spot-welded with a 0.3 mm thick Pt wire, was respectively used as a current collector for the anode and cathode.

The prepared cell, denoted as the $Ni_{1-X}Co_X$ cell, was fired at 850 °C for 30 min to soften Pyrex® glass rings which were used to respectively seal the outer edge of the anode face and that of the cathode each with an alumina tube (15 mm outer diameter and 13 mm inner diameter) edge. The temperature of the cell was decreased to 800 °C and the anode face was exposed to an $H_2$ atmosphere for 2 h to obtain the metallic $Ni_{1-X}Co_X$ prior to the measurement of the single cell performance at 750 °C. The flow rate of $H_2$ (99.9% purity) was fixed at 20 $cm^3$ $min^{-1}$ throughout the measurement. Oxygen gas was fed to the cathode at a flow rate of 20 $cm^3$ $min^{-1}$. The cell performance was evaluated by the current-voltage characteristics. Electrochemical impedance spectra were recorded with a frequency response analyzer (FRA5097, nF) under the open circuit condition. The anodic overvoltage was measured with a digital oscilloscope (TDS2012, Tektronix, Beaverton, OR, USA) by the current interruption method using a current pulse generator (NCPG-101, Nikko Keisoku, Akashi-Shi, Japan). After the cell performance measurement by supplying $H_2$ to the anode, the fuel gas was switched to a mixture of 20 vol%$CH_4$ (99.0% purity) and 80 vol% He (99.995% purity) fed at a total flow rate of 20 $cm^3$ $min^{-1}$. The electrochemical measurement was made between 550 and 750 °C. Subsequently, the operating temperature was kept at 750 °C, and the terminal voltage and impedance spectra were recorded at 80 mA to investigate the performance stability for 60 h. Separately, the symmetrical cells using the same YSZ electrolyte disk sandwiched on both sides with the $Ni_{1-X}Co_X$ (x = 0, 0.25)/YSZ cermet electrodes were prepared under the same method described above. The electrode interface conductivity was determined by the electrochemical impedance spectroscopy in a $CH_4$ atmosphere for the prepared symmetrical cells, which were treated in the same manner as prepared for the anode. The reproducibility of cell performance, including the prolonged stability test was confirmed at least two times for the Ni cell and the $Ni_{0.75}Co_{0.25}$ cell.

Microstructural observations of the $Ni_{1-X}Co_X$/YSZ cermet anodes were made by scanning electron microscopy (SEM, SU8000 and TM3000, Hitachi High-Tech, Tokyo, Japan). The carbon deposition to the surface of the $Ni_{1-X}Co_X$/YSZ cermet anode after the performance stability test supplied with $CH_4$ was investigated with the SU8000 combined with an energy-dispersive X-ray spectrometer (EDX, X-Max$^N$ 80, Oxford Instruments, Abingdon, UK). The $Ni_{1-X}Co_X$O-YSZ powders for phase identification were prepared in the same method as described above and reduced under the same condition to obtain the $Ni_{1-X}Co_X$/YSZ cermet powders. The phase identification of theses powders was made by X-ray diffraction (XRD, HF-2100, Rigaku, Tokyo, Japan) with the monochoromated Cu-K$\alpha$ emission powered at 40 kV and 30 mA.

## 3. Results and Discussion

### 3.1. Phase Identification

Figure 1 shows the XRD patterns of the $Ni_{1-X}Co_X$/YSZ cermet powders. The diffraction peaks both from the $Ni_{1-X}Co_X$ and YSZ phases were indexed as the face-centered cubic (fcc). Increasing the Co content resulted in a decrease in the peak intensities of the $Ni_{1-X}Co_X$ phase with a shift in the peaks to the low diffraction angle side due to the incorporation of Co atoms having the larger atomic size (1.52 Å) than that of Ni (1.49 Å) into the fcc Ni lattice [24]. The observed peak-intensity decrease of the $Ni_{1-X}Co_X$ phase with the increase in the Co content corresponded with the similar decrease of the as-heated $Ni_{1-X}Co_XO$ phase at 1300 °C, as shown in Figure S1. This intensity decrease appears to be connected with an insufficient crystallization of the $Ni_{1-X}Co_XO$ solid solution on heating. In contrast, the peak intensities of the YSZ phase showed an increase in accordance with an appearance of the sharp peak pattern with the increase in the Co content, for example, the 311 reflection exhibiting well resolved $K\alpha_1$ and $K\alpha_2$ peaks, as shown in Figure 1. Heating the dehydrated $Ni_{0.5}Co_{0.5}O$/YSZ precursor at 800 °C led to an appearance of the $Co_3O_4$ phase, as shown in Figure S2. The $Co_3O_4$ phase disappeared after subsequent heating at 1300 °C in air. Since $Co_3O_4$ melts at 895 °C and decomposes into CoO at about 950 °C, $Co_3O_4$ can serve as a sintering aid for the mixed $Ni_{1-X}Co_XO$ and YSZ powders. Although the $Co_3O_4$ phase was not identified for the $Ni_{1-X}Co_XO$-YSZ (x = 0.05, 0.15 and 0.25) precursor powders heated at 800 °C, the changes in the peak shapes and intensities with the Co content suggest that the formation of $Co_3O_4$ as a by-product leads to an enhancement of the sintering of the $Ni_{1-X}Co_XO$-YSZ powders. The observed gradual changes in the diffraction peaks from the $Ni_{1-X}Co_X$ and YSZ phases in the XRD profiles support this.

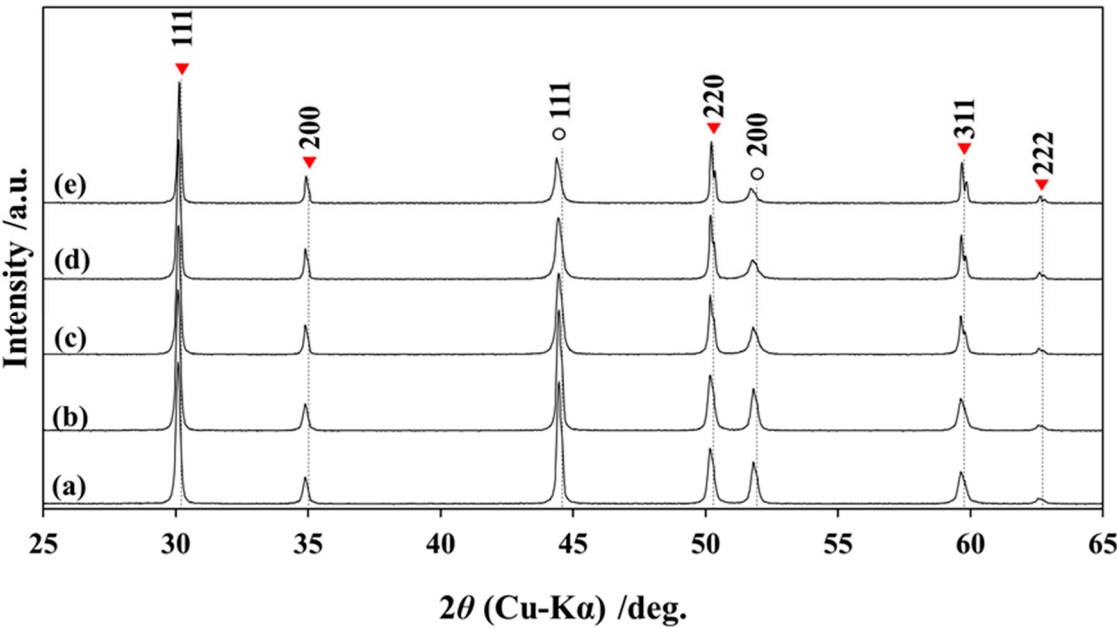

**Figure 1.** XRD patterns of the $Ni_{1-X}Co_X$/ yttria-stabilized zirconia (YSZ) cermet powders prepared by $H_2$ reduction at 800 °C for 2 h (○ : $Ni_{1-X}Co_X$, ▼: YSZ). (**a**) x = 0, (**b**) x = 0.05, (**c**) x = 0.15, (**d**) x = 0.25, and (**e**) x = 0.50.

### 3.2. Microstructure of the Anode

Figure 2a–e show backscattered electron images of fractured cross-sections of the $Ni_{1-X}Co_X$/YSZ cermet anodes. These porous anodes comprise bright dense YSZ grains and dark sponge-like $Ni_{1-X}Co_X$ grains, and both of the grains were well connected by sintering. The average size of the YSZ grains increased with the Co content, which was induced by a promoting effect of the impregnated mixture of the $Ni_{1-X}Co_XO$ and YSZ powders on the sintering in air. This effect corresponds to promoting the

effect of cobalt oxide addition to NiO and YSZ on their sintering [24,25]. Figure 2f shows a cross section of the anode and electrolyte of the $Ni_{0.75}Co_{0.25}$ cell, which revealed that the anode and electrolyte were well bonded and the anode layer thickness was constant.

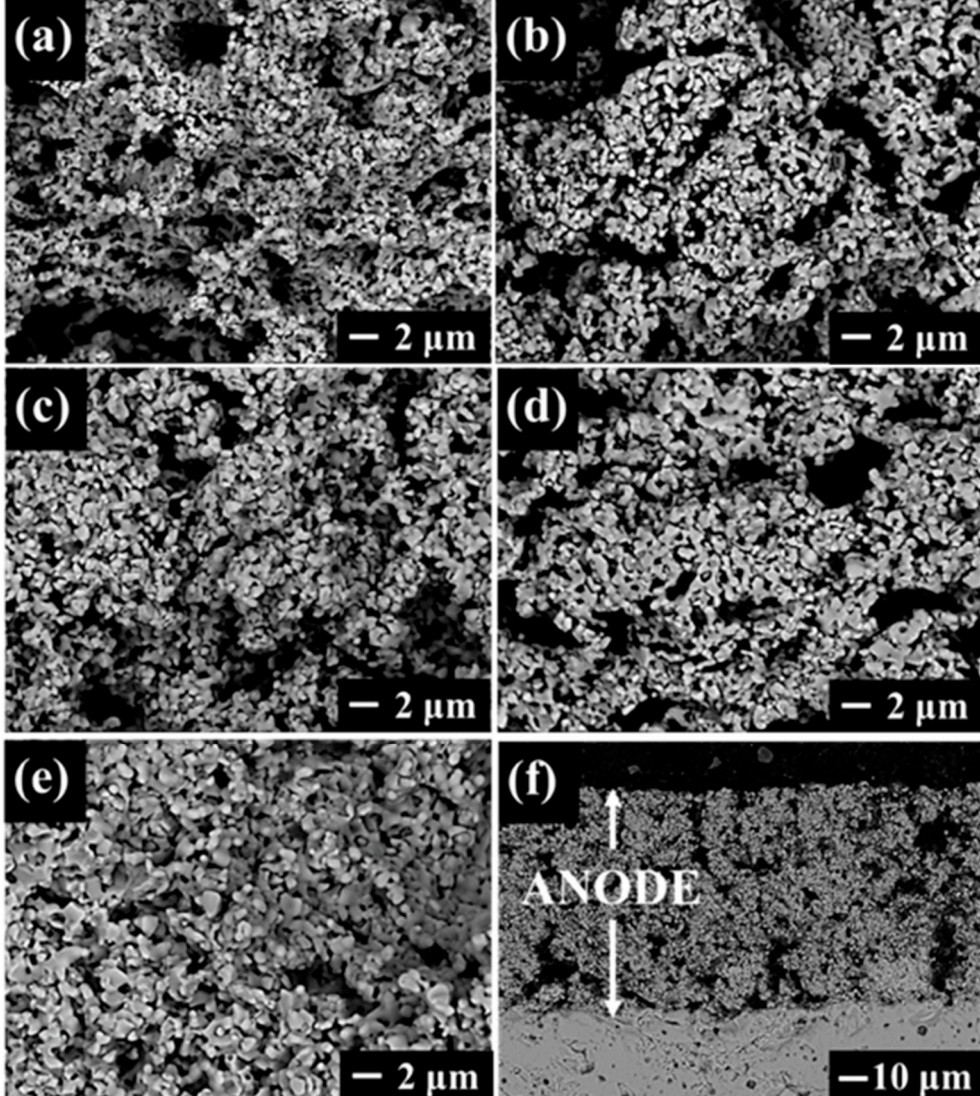

**Figure 2.** Backscattered electron images of the cross sections of the $Ni_{1-X}Co_X$/YSZ cermets: (**a**) x = 0, (**b**) x = 0.05, (**c**) x = 0.15, (**d**) x = 0.25, (**e**) x = 0.50 and (**f**) x = 0.25 with the YSZ electrolyte disk.

### 3.3. Cell Performance for a $H_2$ Supply

Figure 3a shows the cell voltage (*E*) and power density (*P*) as a function of the current density (*J*) of the $Ni_{1-X}Co_X$ cells supplied with $H_2$ at 750 °C. The open circuit voltage (OCV) was 1.4 V, which agreed with the theoretically calculated value provided that the equilibrium partial pressure of oxygen ($P_{O_2}$) is determined by the fixed partial pressure ratio of water vapor pressure ($P_{H2O}$) in the supplied $H_2$ gas and its total pressure ($P_{H2}$) as $P_{H_2O}/P_{H_2}$ = 0.0001. The performance of the $Ni_{1-X}Co_X$ cells was determined by the Co content; the $Ni_{0.75}Co_{0.25}$ cell showed the highest maximum power density of 156 mW/cm$^2$. The performance of the $Ni_{1-X}Co_X$ cells increased in the order, x = 0 < x = 0.05 < x = 0.15, x = 0.5 < x = 0.25. This result implies that the $Ni_{0.75}Co_{0.25}$/YSZ anode has the most optimum porous structure for the transport of supplied $H_2$ to the reaction sites and the lowest electrical resistance due to the well-connected YSZ and $Ni_{0.75}Co_{0.25}$ grains.

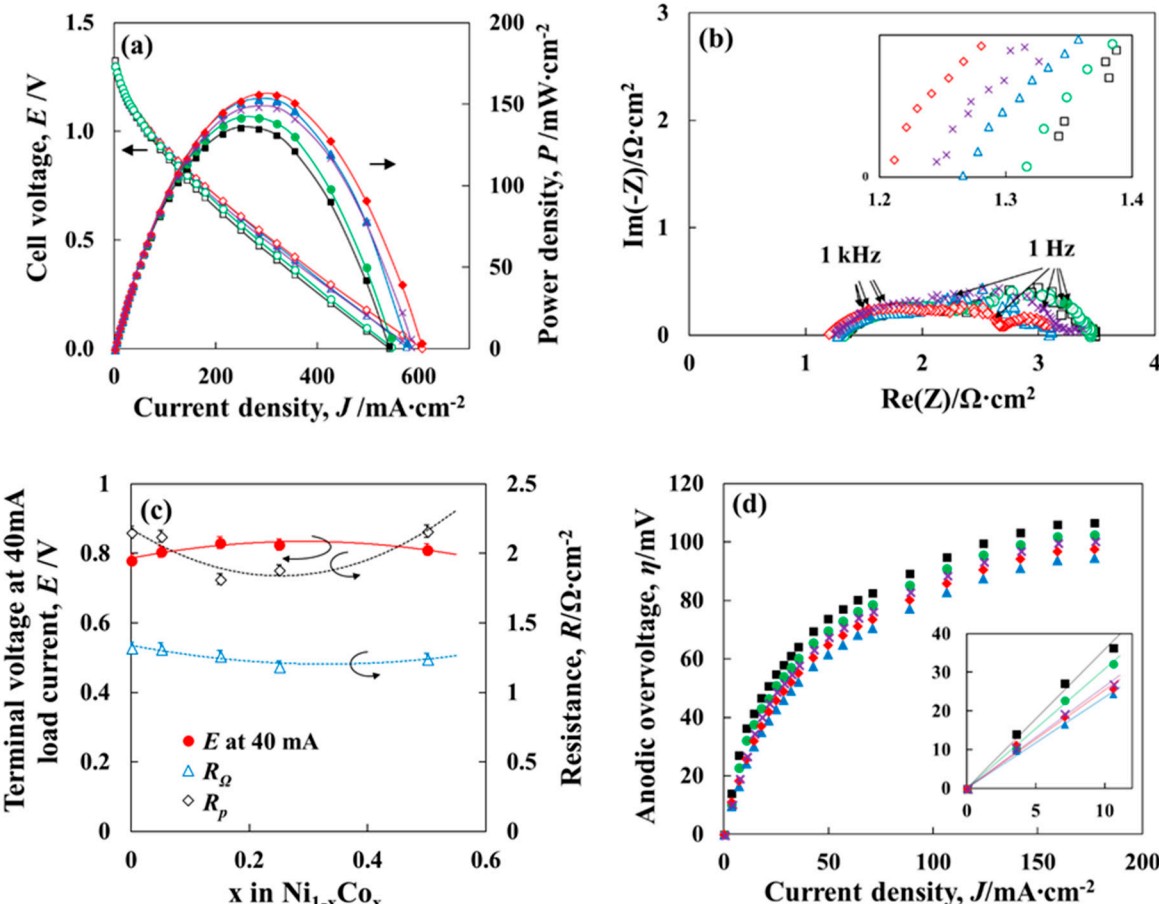

**Figure 3.** (**a**) Current–voltage curves (open symbols) and power densities (closed symbols), (**b**) AC impedance spectra measured at open circuit voltage (OCV), (**c**) the cell voltage at a constant current, the ohmic resistance and the polarization resistance as a function of x, and (**d**) the anodic overvoltages of the $Ni_{1-X}Co_X$ cell at 750 °C for the $H_2$ supply: x = 0 (■,□); x = 0.05 (●,○); x = 0.15 (▲,△); x = 0.25 (◇,◆); x = 0.50 (X). The symbols in Figure 3c are different from Figure 3a,b,d.

Figure 3b shows the electrochemical impedance spectra of the $Ni_{1-X}Co_X$ cells. The real axis intercept at the high frequency region represents the ohmic resistance ($R_\Omega$) of the cell, which includes the overall contact resistance among each of the cell components; the electrode, electrolyte, and current collector. The real axis intercept at the low frequency region represents the total resistance ($R_{tot}$), which includes the ohmic resistance combined with the polarization resistance ($R_p$). The polarization resistance ($R_p$) was determined by subtracting the ohmic resistance from the total resistance. Figure 3c shows that the $Ni_{0.75}Co_{0.25}$ cell showed the lowest $R_\Omega$ and $R_p$ and the highest cell voltage at a constant current, which agreed with the results shown in Figure 3a. Figure 3d shows a behavior of the anodic overvoltage ($\eta$) for the $Ni_{1-X}Co_X$ cells as a function of *J*. At the low current density range, the initial slope of the curve represents the activation overvoltage shown in the inset of Figure 3d, while at the higher current density range (>10 mA cm$^{-2}$), the diffusion overvoltage is dominant in the anodic overvoltage. Both the activation and diffusion overvoltages of the $Ni_{1-X}Co_X$ cells decreased with an increase in the Co content (x) and showed the lowest values at x = 0.15, whereas they turned to increase with x > 0.15. This result indicates that the optimum Co alloying into the Ni matrix provides a high activity for the $H_2$ oxidation and a favorable microstructure as the cermet anode, which accorded with the variation of the impedance spectra of the $Ni_{1-X}Co_X$ cells.

As shown in Figure 2, the grain growth of YSZ particles during sintering occurs significantly with an increase of x, which promoted the grain connectivity between the $Ni_{1-X}Co_X$ metal and YSZ with a compensation for the decrease of their contact area, resulting in an electrical resistance increase.

Venkataramanan et al. reported that a theoretical calculation of an Ni cluster alloyed with 20 mol% Co using the quantum chemical method predicts the chemisorption energy of hydrogen on the metal cluster (0.541 eV/hydrogen atom), which was lower than that on a pure Ni cluster (0.689 eV/hydrogen atom). They also showed that the H–H bond length at a transition state of hydrogen dissociation process of the Ni cluster was extended from 1.534 to 1.621 Å [26]. These results suggest that Co alloying facilitates the dissociation of $H_2$ molecules on the $Ni_{1-X}Co_X$ surface, resulting in promoting the electrochemical oxidation of hydrogen at the $Ni_{1-X}Co_X$/YSZ anode. However, the overvoltage of the $Ni_{1-X}Co_X$ cells increased at x > 0.15, which was caused by a decrease in the TPB length resulting in an increase of the polarization resistance with a decrease of the cell performance. Furthermore, the electrical conductivity of Ni is known to decrease by Co alloying due to electron scattering enhancement [23], which can be a factor for lowering the cell performance.

### 3.4. Cell Performance for a $CH_4$ Supply

Figure 4a shows *J–E* and *J–P* curves of the $Ni_{1-X}Co_X$ cells supplied with $CH_4$ at 750 °C. The performance of the $Ni_{1-X}Co_X$ cell increased in a similar way to that shown in Figure 3a; x = 0 < x = 0.05 < x = 0.5 < x = 0.15 < x = 0.25. The maximum power density of the Ni cell was 94 mW/cm$^2$ and that of the $Ni_{0.75}Co_{0.25}$ cell was 136 mW/cm$^2$; a 45% increase was identified. By comparison, for the $H_2$ supply, the $Ni_{0.75}Co_{0.25}$ cell showed a 17% increase compared with the Ni cell, as shown in Figure 3a. Figure 4b shows the impedance spectra of the $Ni_{1-X}Co_X$ cells, and Figure 4c shows $R_\Omega$ and $R_p$ as a function of x. The $Ni_{0.75}Co_{0.25}$ cell exhibited the lowest $R_\Omega$ and $R_p$ as well as the highest terminal voltage at a constant current. Figure 4d shows $\eta$ as a function of *J*. The inset shows the initial linear slope range indicating the activation overvoltages of the $Ni_{1-X}Co_X$ cells. The Ni cell exhibited the highest $\eta$ and the $Ni_{0.85}Co_{0.15}$ cell exhibited the lowest one. The $Ni_{0.75}Co_{0.25}$ cell also exhibited a low $\eta$. The polarization resistance determined from the initial linear slope for $CH_4$ was found to be four to five times higher than that for $H_2$. This indicates that the electrochemical oxidation of $CH_4$ requires a higher voltage drop than that of $H_2$.

### 3.5. Temperature Dependence of the Anode Interface Conductivity

Figure 5 shows the Arrhenius plot for the apparent electrode interface conductivity, which contains a contribution from the YSZ electrolyte. Since we used YSZ disks with the same size and the polished surfaces, the contribution can be ignored to compare the relative difference in the impedance among the measured samples. The impedance spectra were collected for both the symmetrical Ni and $Ni_{0.75}Co_{0.25}$ cells in a $CH_4$ atmosphere in the temperature range of 550 (or 600)–750 °C. The resistance was determined from the difference between the high-frequency arc and low-frequency arc intercept, as shown in Figures S3 and S4. The reciprocal resistance was regarded as the apparent electrode interface conductivity, which corresponds to the charge transfer occurring at the interface between the $Ni_{1-X}Co_X$ and YSZ grains involving adsorbed $CH_4$ molecules. The activation energies obtained from the Arrhenius plot were 98 kJ mol$^{-1}$ for the Ni cell and 96 kJ mol$^{-1}$ below 650 °C and 39 kJ mol$^{-1}$ above 650 °C for the $Ni_{0.75}Co_{0.25}$ cell. The activation energies for the Ni cell between 600 and 750 °C, and that for the $Ni_{0.75}Co_{0.25}$ cell below 650 °C were almost the same, whereas that for the $Ni_{0.85}Co_{0.25}$ cell above 650 °C exhibited a much lower energy. Thermal equilibrium of the $CH_4$ decomposition reaction is known to be product (C and $H_2$) favorable at temperatures greater than about 700 °C. A possible interpretation of the above result is that the thermal dissociation of $CH_4$ on the $Ni_{0.85}Co_{0.25}$ surface is more facile and favorable for the following electrochemical steps of the partial or complete oxidation of $CH_4$ accompanying the charge transfer from the oxide ions to electrons at the interface between $Ni_{1-X}Co_X$ and YSZ and desorption of the products, CO, $H_2$, $CO_2$, and $H_2O$. Li et al., reported a computation calculation of the activation energy of the $CH_4$ dissociation on Ni and NiCo. The calculation revealed that the activation energy (*Ea*) of $CH_4$ dissociation on the Ni(111) surface (1.36 eV (= 131 kJ mol$^{-1}$)) was higher than that of the NiCo(111) surface (1.29 eV (= 124 kJ mol$^{-1}$)) and the Co(111) (1.25 eV(= 121 kJ mol$^{-1}$)) [27,28]. Although their calculation was

done on the well-defined single crystal surface, lowering the *Ea* by Co alloying into the Ni matrix can promote the rate of $CH_4$ dissociation and result in reducing the anodic polarization resistance for the electrochemical oxidation of $CH_4$, provided that the rate-determining step of the anodic oxidation of $CH_4$ is the dissociation of the adsorbed $CH_4$. The Co alloying seems to be effective to control the well-known strong adsorptive property of Ni for CO; the suppression of CO poisoning at the Ni/YSZ-based cermet anode is expected to enhance the cell performance for the direct oxidation of $CH_4$. An enhancement of the cell performance was also reported for the direct oxidation of $CH_4$ by using a $Ni_{0.5}Co_{0.5}$/YSZ anode combined with an ScSZ electrolyte [29]. An appropriate amount of Co alloying in the Ni matrix was found to be effective to reduce the anodic activation overvoltage above about 700 °C.

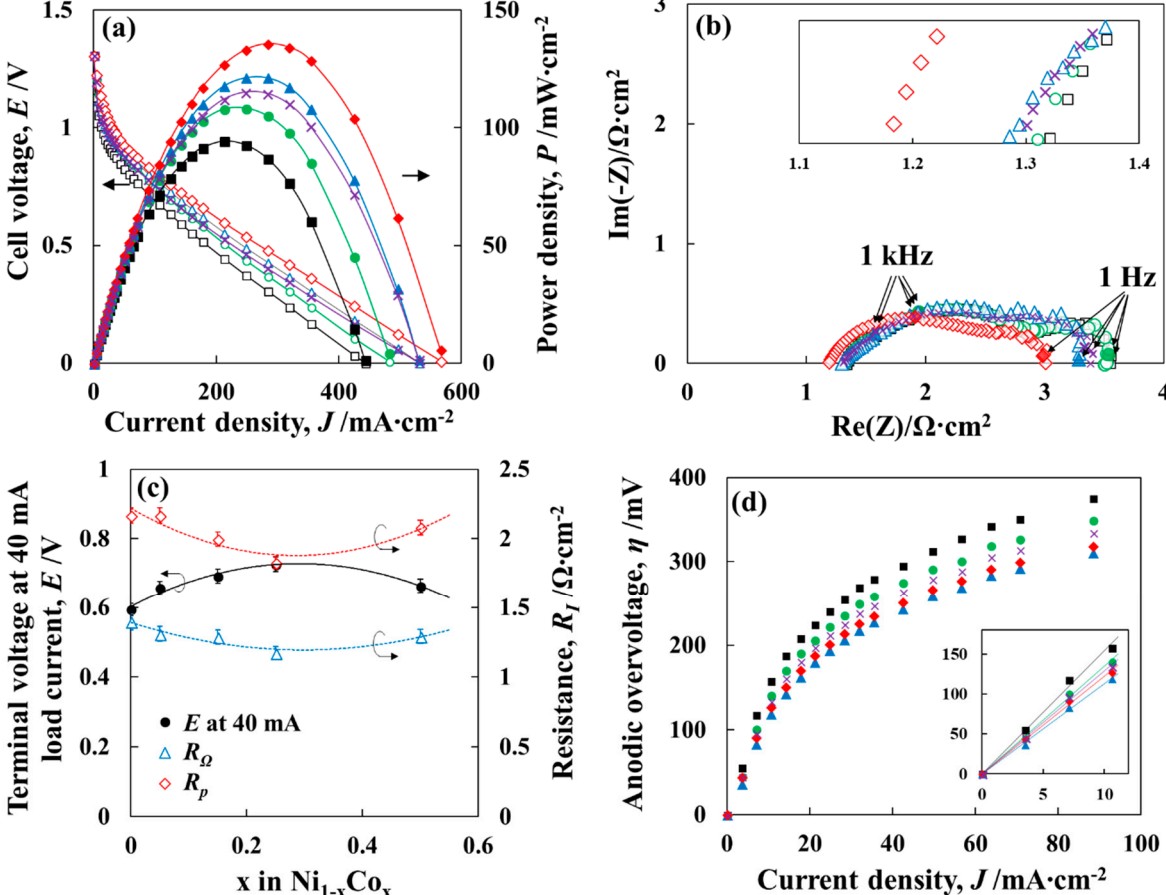

**Figure 4.** (**a**) Current–voltage curves (open symbols) and power densities (closed symbols), (**b**) electrochemical impedance spectra measured at OCV, (**c**) cell voltage at a constant current, ohmic resistance and the polarization resistance as a function of x, and (**d**) the anodic overvoltages of the $Ni_{1-X}Co_X$ cell at 750 °C for the $CH_4$ supply: x = 0 (■,□); x = 0.05 (●,○); x = 0.15 (▲,△); x = 0.25 (◆,◇); x = 0.5 (X). The symbols in Figure 4c are different from Figure 4a,b,d.

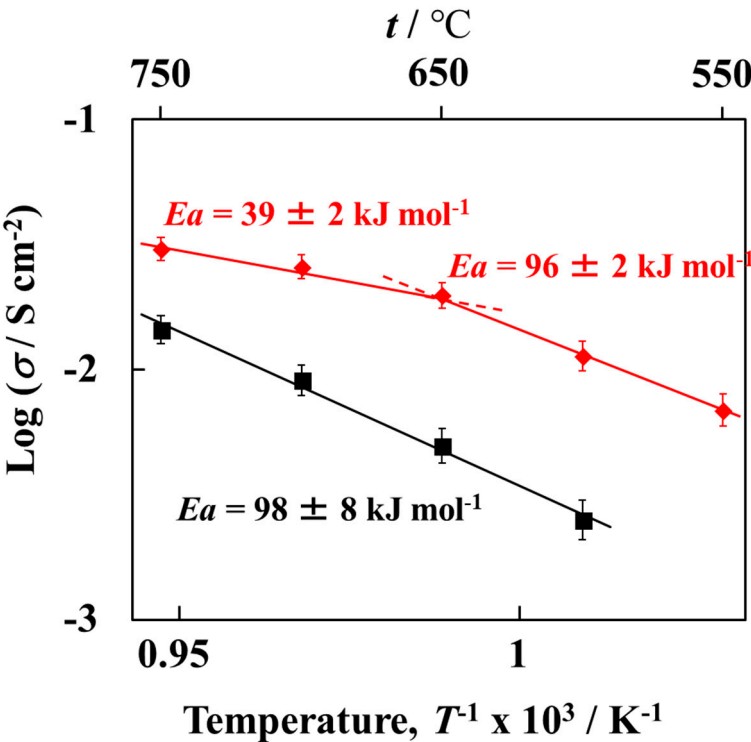

**Figure 5.** Arrhenius plots for the apparent electrode interface conductivity of the Ni/YSZ|YSZ|Ni/YSZ (■) and $Ni_{0.75}Co_{0.25}$/YSZ|YSZ|$Ni_{0.75}Co_{0.25}$/YSZ (◆) cells in $CH_4$.

### 3.6. Prolonged Stability in $CH_4$

The $Ni_{0.85}Co_{0.15}$ and $Ni_{0.75}Co_{0.25}$ cells were chosen to investigate the prolonged cell performance stability for $CH_4$ to compare with the Ni cell, which is based on the results that the $Ni_{0.85}Co_{0.15}$ cell exhibited the lowest anodic overvoltage and the $Ni_{0.75}Co_{0.25}$ cell exhibited the lowest ohmic resistance. Figure 6a shows changes in the total resistance and cell voltage at a constant current of 80 mA at 750 °C as a function of operating time for the Ni, $Ni_{0.85}Co_{0.15}$, and $Ni_{0.75}Co_{0.25}$ cells. The cell voltage of the Ni cell significantly decreased from 0.62 to 0.45 V (about 27% decrease) after operation for 60 h. This decrease corresponded to the increase of the total resistance (= $R_p + R_\Omega$) determined from the impedance spectra under the OCV condition, which is shown in Figure 6b. In contrast, the cell voltage of the $Ni_{0.85}Co_{0.15}$ cell changed from 0.69 to 0.67 V (about 3% decrease) and that of the $Ni_{0.75}Co_{0.25}$ cell changed from 0.73 to 0.70 V (about 4% decrease); the cell voltage drop was almost insignificant at least up to 60 h. Figure 6b shows a comparison of the impedance spectra before and after the prolonged stability test. The Ni cell showed a larger increase in the impedance arc with a shift to the high value of the real axis, Re(Z), than the $Ni_{0.85}Co_{0.15}$ and $Ni_{0.75}Co_{0.25}$ cells with slight shifts of their impedance arcs. The significant increase of $R_p + R_\Omega$ for the Ni cell is probably caused by the thermal grain growth of Ni, which reduces the contract area between the Ni and YSZ, and the deactivation of the nickel surface with a concurrent carbon deposition. On the other hand, the increases in $R_p + R_\Omega$ for the $Ni_{0.85}Co_{0.15}$ and $Ni_{0.75}Co_{0.25}$ cells were small, which indicates no significant grain growth of Ni as well as the sufficiently active surface state against an exposure of $CH_4$.

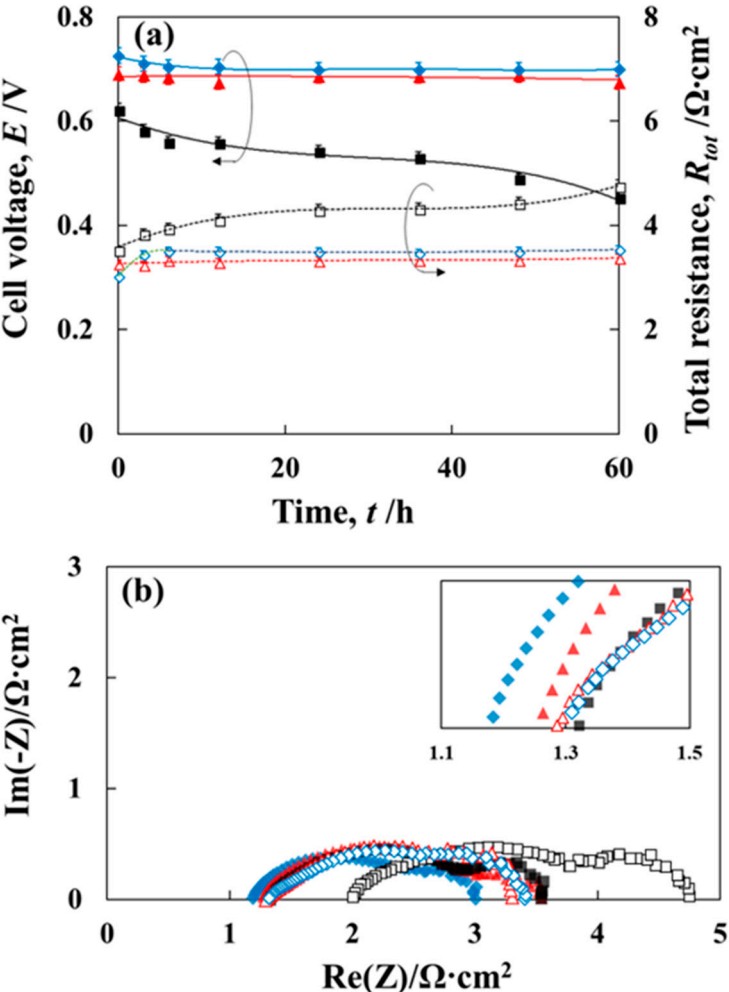

**Figure 6. (a)** The cell voltage (closed symbol) stability and the total resistance (open symbols) for the $Ni_{1-X}Co_X$ cells with time at a constant current at 750 °C for the $CH_4$ supply: x = 0 (■,□); x = 0.15 (▲,△); x = 0.25 (◆,◇). **(b)** The electrochemical impedance spectra measured at OCV before and after the prolonged test shown in (a): x = 0 (■:0 h, □:60 h); x = 0.15 (▲:0 h, △: 60 h); x = 0.25 (◆: 0 h, ◇: 60 h).

After this prolonged stability test, the anode surface morphologies of the Ni and $Ni_{0.85}Co_{0.15}$ cells were investigated by SEM. Elemental analysis was also made for these anode surfaces. Figure 7a–d show the surface morphologies and corresponding area analysis by EDX. The carbon peak that appeared in the Ni cell is much stronger than that in the $Ni_{0.85}Co_{0.15}$ cell. This result indicates that the proper Co alloying for the Ni-based/YSZ cermet anode by the impregnation treatment delivers excellent cell performance for the direct supply of $CH_4$ because of the favorable cermet microstructure having low electrical resistance and the surface state resistant to the deactivation by adsorbed $CH_4$. The formation of carbon on an SOFC anode directly exposed to $CH_4$ can easily occur via the pyrolysis reaction (Equation (1)) at temperatures higher than about 700 °C. The thermally dissociate carbon can be consumed by the electrochemical oxidation (Equation (2)) with the supply of $O_2^-$ ions through the electrolyte when the product of adsorbed CO is removed from the Ni-based metal surface.

$$CH_4 \rightarrow C + 2H_2 \tag{1}$$

$$C + O^{2-} \rightarrow CO + 2e \tag{2}$$

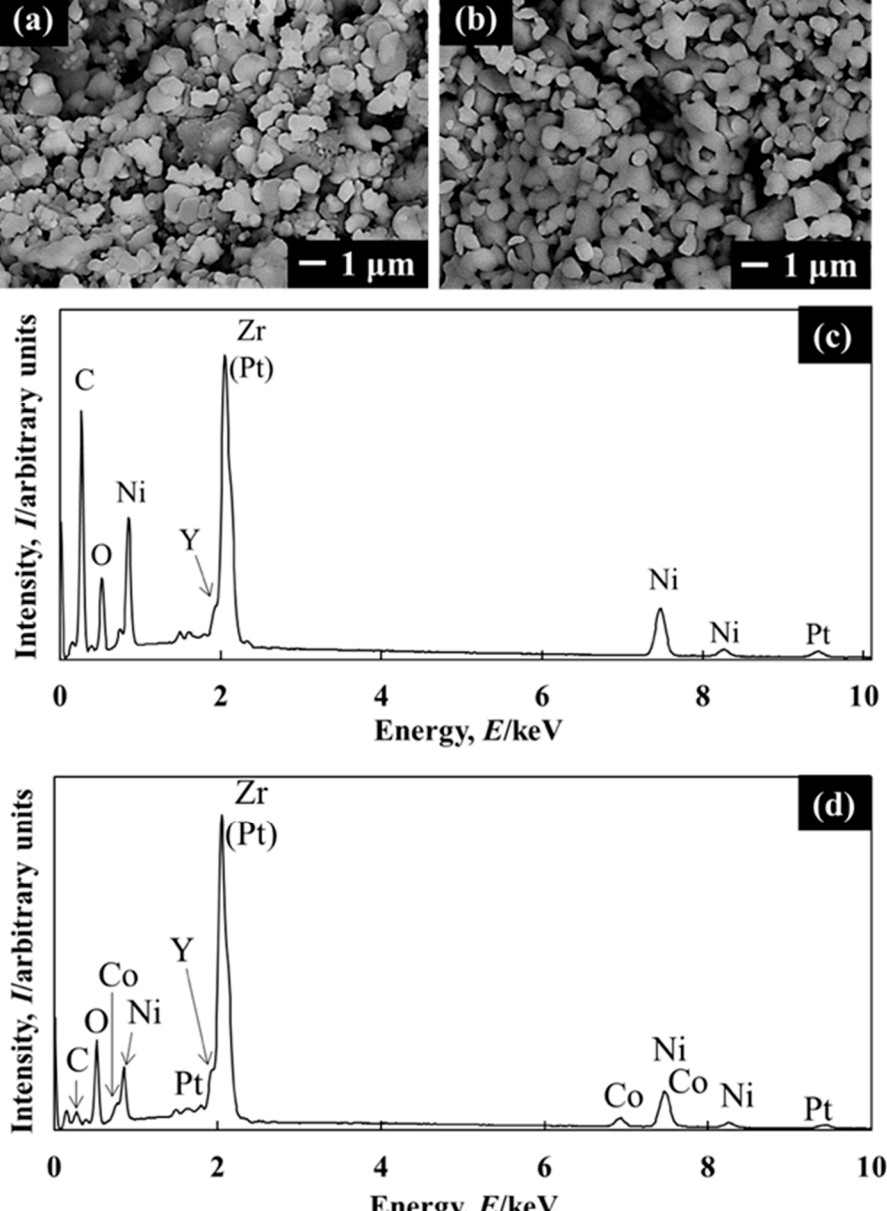

**Figure 7.** SEM micrographs of the anode surface for (**a**) the Ni cell (**b**) the $Ni_{0.85}Co_{0.15}$ cell, and the corresponding EDX spectra of (**c**) the Ni cell (**d**) the $Ni_{0.85}Co_{0.15}$ cell after the prolonged test at 750 °C for 60 h in $CH_4$.

Since the desorption energy of CO from the Co metal surface (161 kJ $mol^{-1}$) was found to be lower than that from the Ni metal surface (173 kJ $mol^{-1}$) [30], the CO desorption from the Co-alloyed Ni/YSZ cermet can be more facile than from the Ni/YSZ anode, which results in a decrease in carbon deposition on the anode surface and provides stable electrochemical performance.

## 4. Conclusions

A modified porous anode of the $Ni_{1-X}Co_X$/YSZ (yttria-stabilized zirconia) cermet was prepared by an impregnation method and was applied to the SOFC operation with direct $CH_4$ supply. The cermet microstructure was improved owing to the promoting effect of the $Co_3O_4$ phase as a sintering aid on the fabrication of the Ni-based cermet anode. The appropriate Co alloying content, x = 0.15–0.25, was found to increase the cell performance with an excellent prolonged cell performance stability as

well as minimizing the carbon deposition thermally from the dissociated $CH_4$. Co was incorporated into Ni and formed a solid solution of $Ni_{1-X}Co_X$ alloy connected with the YSZ as the cermet anode. The performance enhancement was found to be more significant for $CH_4$ than that for $H_2$ compared with the performance of the cell using the Ni/YSZ cermet anode (an improvement of 17% in $H_2$ fuel and 45% in $CH_4$ fuel for x = 0.25). This enhancement effect for the direct $CH_4$ supply was probably caused by the optimum porous microstructure of the cermet anode with the low anodic polarization resistance, which is specific to an operating temperature higher than about 700 °C.

**Supplementary Materials:** The following are available online at http://www.mdpi.com/2571-6131/3/1/12/s1, Figure S1: XRD patterns of the $Ni_{1-X}Co_XO$-YSZ powders prepared by sintering at 1300 °C for 3 h in air. Figure S2: XRD patterns of the Ni1-XCoXO-YSZ powders prepared by calcination at 800 °C for 5 h in air. Figure S3: Impedance spectra of symmetric cell measured at 600–800 °C under H2 atmosphere. Figure S4: Impedance spectra of symmetric cell measured at 600–800 °C under CH4 atmosphere.

**Author Contributions:** Conceptualization, K.S. and S.C.; methodology and data curation, N.W. and K.Y.; data analysis, methodology and discussion, M.N.; validation, M.N. and K.S.; writing—original draft preparation, N.W.; writing—review and editing, K.S.; project administration and funding acquisition, K.S. All authors have read and agreed to the published version of the manuscript.

**Funding:** This work was supported by JSPS Grant-in-Aid for Scientific Research B, grant number 24360304.

**Acknowledgments:** One of the authors (N.W.) would like to acknowledge the financial support by a scholarship in the form of a grant from the Pacific Rim Green Innovation Hub Project, Nagaoka University of Technology. Some of the analytical instruments were provided by the advanced methane research utilization project supported by Nagaoka University of Technology.

**Conflicts of Interest:** The authors declare no conflict of interest.

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
