# Peer review of "Improved Electrochemical Properties of an Ni-Based YSZ Cermet Anode for the Direct Supply of Methane by Co Alloying with an Impregnation Method"

_ceramics, doi:10.3390/ceramics3010012_

Round 1

Reviewer 1 Report

Conclusions should be presented in more detaile.

Reviewer 2 Report

This manuscript describes the behaviour of the solid solution of a Ni1-xCox alloy as a cermet anode for direct supply of methane in solid oxide fuel cells (SOFC). I have found the article very well written, with good experimental details, clear results and figures as well as very interesting discussions of these results with a nice comparison of the behaviour either in H2 or CH4 as gas supply. I believe this manuscript provides very valuable information for the SOFC community and I recommend the publication as it is. 

Reviewer 3 Report

This study investigates the effects of Ni-Co solid solution at the anode of SOFCs on electrochemical performance using H2 or CH4 fuels. The experiments have been well-designed and systematically carried out to address how the composition of alloy affects microstructure and electrode reaction. This manuscript is well-configured with acceptable logical progression, however, there exist several minor points that should be considered/corrected in the revised manuscript.  

1) Phase identification part: this part is a bit confusing. If I understand correctly, XRD analysis was performed for the cermet powders prepared by three different conditions: i) H2 reduction at 800C, ii) H2 reduction at 1300C, and iii) calcination at 800C in air followed by 1300C in air. If the authors can describe the detail of heat-treatment conditions and the purpose using such conditions in experimental part or discussion part, it would be better to make the readers grasp the concept quickly. Also, the caption of Figure S2 should be corrected. In current version, caption of Figure S2 is identical with that of Figure S1.

2) In Fig. 3(c) and 4(c), x=0.25 composition shows the lowest ohmic resistance. As the authors suggested and clearly showed in microstructural images, higher Co content accelerates sintering of YSZ in anode cermet, and in turn, better connectivity between the anode and electrolyte can be achieved. Is that because higher Co content decreases electrical conductivity of Ni-Co alloy? It is a bit unclear why x=0.50 sample exhibits higher ohmic resistance than x=0.25.

3) several typos:

P3 line 138: coressponded -> corresponded

   Line 143: N0.5 -> Ni0.5

P4 line 146, 147, 149: N1-x -> Ni1-x

   Line 147: x=0.10 -> x=0.15

Fig. 2 caption: Co0x -> Cox

Fig. 5: cations for red and black symbols are missing

Fig. 6: color of symbol for x=0.25 should be blue, not red.
